# Balancing Accuracy and Speed in Gaze-Touch Grid Menu Selection in AR via Mapping Sub-Menus to a Hand-Held Device

**DOI:** 10.3390/s23239587

**Published:** 2023-12-03

**Authors:** Yang Tian, Yulin Zheng, Shengdong Zhao, Xiaojuan Ma, Yunhai Wang

**Affiliations:** 1Guangxi Key Laboratory of Multimedia Communications and Network Technology, School of Computer, Electronics and Information, Guangxi University, Nanning 530004, China; yulinz@st.gxu.edu.cn; 2Department of Computer Science, National University of Singapore, Singapore 119077, Singapore; zhaosd@comp.nus.edu.sg; 3Department of Computer Science and Engineering, Hong Kong University of Science and Technology, Hong Kong 999077, China; mxj@cse.ust.hk; 4School of Computer Science and Technology, Shandong University, Qingdao 266237, China; cloudseawang@gmail.com

**Keywords:** gaze and touch, multi-modal interaction, sub-menu

## Abstract

Eye gaze can be a potentially fast and ergonomic method for target selection in augmented reality (AR). However, the eye-tracking accuracy of current consumer-level AR systems is limited. While state-of-the-art AR target selection techniques based on eye gaze and touch (gaze-touch), which follow the “eye gaze pre-selects, touch refines and confirms” mechanism, can significantly enhance selection accuracy, their selection speeds are usually compromised. To balance accuracy and speed in gaze-touch grid menu selection in AR, we propose the Hand-Held Sub-Menu (HHSM) technique.tou HHSM divides a grid menu into several sub-menus and maps the sub-menu pointed to by eye gaze onto the touchscreen of a hand-held device. To select a target item, the user first selects the sub-menu containing it via eye gaze and then confirms the selection on the touchscreen via a single touch action. We derived the HHSM technique’s design space and investigated it through a series of empirical studies. Through an empirical study involving 24 participants recruited from a local university, we found that HHSM can effectively balance accuracy and speed in gaze-touch grid menu selection in AR. The error rate was approximately 2%, and the completion time per selection was around 0.93 s when participants used two thumbs to interact with the touchscreen, and approximately 1.1 s when they used only one finger.

## 1. Introduction

Eye gaze is a promising interaction modality for target selection in augmented reality (AR) due to its potential speed and ergonomic advantages [1]. However, eye gaze is widely known to be unnatural and inaccurate due to various factors such as the Midas problem [2] and measurement errors of the eye-tracking systems [1,3,4], etc.

To address the Midas problem, a secondary interaction modality for confirming the selection of the target is often used in conjunction with eye gaze pointing, in addition to the naive method, “dwell” [5]. The secondary modality may include using two fingers to make an air-tap gesture, pressing a button on a device, or other methods. To compensate for the low accuracy of eye gaze, techniques such as those proposed in Look&Touch and Pinpointing [1,3] introduced a refinement phase for fine-tuning the position of the cursor associated with eye gaze (gaze-driven cursor) between the gaze pointing and selection confirmation phases. After pre-selecting a target item via gaze pointing under these techniques, users can utilize touch input to refine the gaze-driven cursor’s position and then confirm the selection. However, an additional refinement phase typically results in not only better target selection accuracy but also longer selection times. For instance, the selection times of all Pinpointing techniques [1] exceed 2.5 s.

The aforementioned gaze-touch techniques [1,3] follow the “eye gaze pre-selects, touch refines and confirms” mechanism and are specifically designed for selecting a target from a set of randomly distributed candidate items. When candidate items are positioned regularly, such as in a grid or radial layout, mechanisms that utilize these regular layouts can better balance accuracy and speed in gaze-touch menu selection in AR. Inspired by the GAT technique [4], which divides a virtual keyboard into several sub-keyboards with radially positioned keys, we propose the Hand-Held Sub-Menu technique (HHSM) for accurate and fast gaze-touch target selection in various grid menus in AR, consisting of icons of applications, folders, files, photos, etc. As illustrated in Figure 1, HHSM divides a grid menu into several sub-menus, with the sub-menu pointed to by eye gaze mapped to the touchscreen of a hand-held device, such as a cellphone or tablet. To select a target item, users first use eye gaze to select the sub-menu that contains the target item, and then confirm the selection of the target item via a single touch action on the touchscreen. HHSM has the potential to balance accuracy and speed in gaze-touch grid menu selection in AR, as it can provide large candidate items (sub-menus) for gaze pointing and a limited number of regularly distributed distractors for easy selection confirmation via touch input.

In this work, we first derived the design space of HHSM, which includes three dimensions: sub-menu size, holding gesture (the way a user holds the device), and the touch input technique for selection confirmation. Next, we conducted a pilot study to finalize the levels of the sub-menu size dimension. As the design space yields numerous instances of HHSM (9 sub-menu sizes × 3 holding gestures × 2 touch input techniques = 54 instances), we conducted another pilot study to identify the optimal instances for the three holding gestures, respectively. Lastly, we conducted a user study to evaluate the HHSM technique by comparing it with three state-of-the-art gaze-touch menu selection techniques: *MAGIC touch* [3], *MAGIC tab* [3], and *Eye + Device* [1] under the three holding gestures, respectively. The research process described above is also illustrated in Figure 2. The main findings of the user study are as follows:

There was no significant difference in selection accuracy between HHSM, *MAGIC touch*, *MAGIC tab*, and *Eye + Device*. Their error rates were approximately 2% (∼1 error in 36 selection trials).Under all three holding gestures, the completion times per selection under HHSM were significantly shorter than those under *MAGIC touch*, *MAGIC tab*, and *Eye + Device*.

## 2. Related Work

In this section, we discuss the following two areas that are related to our research.

### 2.1. Incorporating a Mobile Device into VR/AR

A mobile device can serve as a controller in VR/AR to support various interactions, including object selection [6,7,8] and manipulation [9], navigation [10], object registration [11], menu interaction [12,13], etc. Budhiraja et al. [6] found that the performance of virtual object selection in AR was better under the condition where cursors were controlled via touch gestures on a touchscreen device than under the condition where cursors were controlled via device rotation. Pietroszek et al. [8] proposed the Tiltcasting metaphor, i.e., a virtual half-transparent plane in VR relatively static to a cellphone. The virtual objects above the plane were invisible. To select a target, the user first rotated the cellphone to make the plane intersect with the target and then confirmed selection via touch input on the cellphone touchscreen. GyroWand [7] uses a state machine to determine the direction of a ray for object selection cast from a non-hand body location (e.g., chin) into the AR scene. It interprets the relative rotation of a handheld controller (e.g., a cellphone) to achieve this. Ha et al. [9] allow users to rotate and translate virtual objects by rotating a cellphone and sliding their finger on the cellphone touchscreen, respectively. Liang et al. [10] enable users to navigate in VR by adjusting their viewpoint and direction by sliding a finger on the cellphone touchscreen and tilting the cellphone, respectively. Ro et al. [11] enable users to adjust the length of a stylus cast from a cellphone in AR. This allows users to specify (register) the 3D position of a virtual object attached to the stylus tip. Henderson et al. [12] developed an AR tutorial system for tank maintenance. It allows users to start and pause tutorial animations played in AR by tapping the buttons on the touchscreen of a cellphone attached to the user’s wrist. Handymenu [13] divides a cellphone touchscreen into two areas for virtual target selection and menu interaction, respectively.

There is also work aimed at overcoming the size limitation of mobile devices using AR head-mounted displays. VESAD [14] extended a cellphone touchscreen with a virtual surface co-planar with it by utilizing the front-facing camera of an AR headset to track Aruco markers attached to the cellphone. Additionally, Grubert et al. [15] proposed MultiFi, which combined the strengths of an AR headset (i.e., a bigger FOV) and mobile devices (i.e., high resolution, touch input, etc.) to support seamless interaction with widgets distributed over these devices.

Another body of work [16,17] utilized the front-facing camera of a cellphone to track physical props or users’ fingers. TrackCap [16] turns a cellphone into a precise 6DOF tracker that can be attached to a physical prop such as a toy gun. To achieve this, TrackCap attaches a planar target to the VR/AR headset and calculates the pose of the cellphone relative to the target captured by the cellphone’s front-facing camera. Phonetroller [17] mounts a mirror above a cellphone touchscreen, enabling the front-facing camera of a cellphone to capture the user’s thumbs between the mirror and the touchscreen. This setup helps create semi-transparent overlays of thumb shadows in VR and track positions of thumb tips relative to the corners of the touchscreen.

### 2.2. Bimodal Target Selection Involving Eye Gaze

Since eye gaze can be unnatural and inaccurate due to issues such as the Midas problem [2], limitations in the physiological nature of our eyes, and measurement errors of eye tracking systems [1,3,4], researchers have focused on overcoming these challenges by designing bimodal target selection techniques that incorporate eye gaze.

The Midas problem refers to that everywhere one looks, another unintentional command is activated [2]. The conventional solution to the Midas problem, as proposed by Jacob et al. [2], is to use dwell time to indicate the user’s intent of selection confirmation. However, the combination of gaze pointing and dwell confirmation is generally considered to be slow and uncomfortable, as reported in a recent study by Kumar et al. [5]. Sidenmark et al. [18] proposed using the distinction between gaze shift performed only by the eyes and gaze shift supported by both eye and head movements to enable hover interaction and visual exploration around pre-selected targets. Lystbæk et al. [19] proposed the concept of gaze-hand alignment, which involves using gaze for pre-selection and then aligning a finger or a hand-controlled cursor with the user’s line of sight for selection confirmation. However, experimental results showed that gaze-hand alignment techniques did not outperform a baseline condition where the pinch gesture was used for selection confirmation in conjunction with gaze. To enable users to enter a word or personal identification number by glancing at virtual keyboard buttons, Hummer [20], TAGSwipe [5], and TouchGazePath [21] require users to indicate their intent to enter through humming or touch input.

The coarse-to-fine strategy is commonly adopted for target selection [22,23,24]. Similarly, to improve the accuracy of bimodal target selection that involves eye gaze, researchers often add a refinement phase between the gaze pointing and selection confirmation phases. Stellmach et al. [3] proposed a set of refinement techniques called Look&Touch, which effectively enhance the accuracy of basic gaze-touch target selection. The Look&Touch techniques include (i) *MAGIC touch*, which involves tapping/swiping on a touchscreen for fine absolute/relative cursor positioning; (ii) *MAGIC tab*, which involves sliding on a touchscreen to cycle through a list of targets close to the gaze point; (iii) eye-slaved zoom lens, which always follows the gaze point; (iv) semi-fixed zoom lens, which does not follow the gaze point until the user looks beyond its boundary. Selection confirmation is achieved by tapping a button on the touchscreen device. Stellmach et al. [25] further proposed TouchGP and HdGP mechanisms extended from *MAGIC touch* and the aforementioned zoom lenses to support fluently transitioning between roughly and precisely selecting, positioning, and manipulating objects. Another refinement technique is the pinpointing technique [1], which involves a primary pointing motion (i.e., head pointing or gaze pointing) and a secondary refinement motion (i.e., rotation of head/handheld device or hand translation). An air-tap down gesture or holding a button down starts the refinement motion, while the completion action of the refinement motion (i.e., an air-tap up gesture or releasing the button) also confirms selection. Experiment results showed that secondary refinement techniques reduced the mean error angle of the basic gaze-touch target selection from $2.42^{\circ }$ to below $0.5^{\circ }$ at the cost of increasing the mean task completion time from 1.4 s to above 2.5 s.

Unlike prior work [1,3] that utilized a refinement phase, Ahn et al.’s Gaze-Assisted Typing (GAT) technique [4] divides the virtual keyboard into sub-keyboards and maps the one pointed at by eye gaze to a temple of a Google Glass. The keys in each sub-keyboard are positioned radially, with one key center-positioned and others surrounding it. To select a target key, the user first looks at the sub-keyboard containing it. If the target key is center-positioned, the user taps on the temple to confirm its selection. If not, the user uses a finger to swipe along the direction from the center-positioned key to it. However, GAT is designed for text entry using a standalone hardware setup. Thus, the related study’s results may not apply to other tasks and hardware setups. In this study, we focus on another fundamental task in AR, menu selection, and use a common hybrid hardware setup that combines an AR glass and a cellphone [1,3] to investigate the effects of mapping sub-menus to a tangible proxy. A tangible proxy [26,27] typically refers to a physical object or representation used to represent or stand in for something that is abstract or not physically present. In the context of the HHSM technique, the touchscreen of the cellphone serves as the tangible proxy for the sub-menu.

## 3. Design Space of the Hand-Held Sub-Menu Technique

By using the touchscreen of a hand-held device as a tangible proxy for AR sub-menus, we consider the design space for the Hand-Held Sub-Menu (HHSM) technique, which comprises three dimensions: (i) the size of a grid sub-menu (**Size**); (ii) the way a user holds the hand-held device (**Hold**); and (iii) the touch technique used to confirm the selection of a target item in a sub-menu (**Touch**).

**Size.** The larger the sub-menus, the more relaxed the requirement for the accuracy of eye tracking becomes. However, there are more distractors for selection confirmation via touch input on the tangible proxy. Therefore, the size of sub-menus determines whether we can effectively balance providing large targets (sub-menus) for eye gaze and keeping a small number of distractors distributed regularly for selection confirmation via touch input. We use “$n_{1}$R$n_{2}$C” to represent a sub-menu with $n_{1}$ × $n_{2}$ items arranged in a $n_{1}$ × $n_{2}$ grid. For example, in the case of 3R4C, it signifies that the sub-menu comprises twelve items arranged in a 3 × 4 grid. We first consider the isotropic sizes that may be helpful to achieve the above balance isotropically: 1R1C, 2R2C, 3R3C, 4R4C, 5R5C, and so on. Particularly, 1R1C is the baseline level where the AR menu is not divided into sub-menus. Then we sample several sizes with item numbers that are between those of every two isotropic sizes. The final levels of the size dimension are: 1R1C, 1R2C, 2R1C, 1R3C, 3R1C, 2R2C, 2R3C, 3R2C, 3R3C, 3R4C, 4R3C, 4R4C, 4R5C, 5R4C, 5R5C, … See Figure 3a for the first nine sizes.

**Hold.** Users can hold a mobile device using different gestures based on usage scenarios or personal preferences. In cases where users have unilateral upper limb disability or have one hand occupied, such as when they are holding a handrail on a bus or subway, or carrying a shopping bag, they can only use a single hand to hold the device and interact with its touchscreen (see Figure 3b). When users have two hands available, they can use one hand to hold the device and use one finger of the other hand to interact with the touchscreen (see Figure 3c). Alternatively, they can hold the device with both hands by grasping the two short edges and use their thumbs to interact with the touchscreen (see Figure 3d). In summary, the different levels of the hold dimension are Single Hand (SH), One hand holds and the Other interacts (OO), and Two Hands (TH).

**Touch.** Users can use different types of touch inputs to confirm the selection of a target in an AR sub-menu. The first technique is the tap method, which involves tapping the counterpart of the target on the cellphone touchscreen; see Figure 3f. The second technique is the swipe (+ tap) method. If the sub-menu contains a target positioned at the center, the user should use swipe + tap; see Figure 3g. To select a target that is not center-positioned, the user should swipe on the touchscreen in the direction from the sub-menu center to the target. To select the center-positioned target, the user can tap anywhere on the touchscreen. If the sub-menu does not have a center-positioned target, the user can use swipe only to select any target in the sub-menu.

## 4. Pilot Study on Sub-Menu Size

The more complex (larger) the sub-menu, the more difficult it is to confirm target selection accurately. If the swipe (+ tap) technique is used for selection confirmation, the most complex sub-menu that can be used is 3R3C. The reason is that more complex layouts, such as 4R3C, introduce multiple items in certain swiping directions (see an illustration in Figure 4a). However, if the tap technique is used, it is not easy to determine the most complex sub-menu that still allows for acceptable selection confirmation accuracy. Hence, we aimed to identify the most complex sub-menu that still allows for acceptable selection confirmation accuracy when the tap technique is used in this pilot study.

**Participants**. We randomly recruited five male right-handed participants (aged 19–26 years old) from a local university. They are all right-handed and two had AR experience.

**Apparatus.** Our hardware setup consisted of a Microsoft HoloLens 2 AR headset and a MI 10 cellphone with a 162.6 mm × 74.8 mm size. The cellphone served as the hand-held device. The system established a wireless network connection via WiFi between the AR headset and the cellphone, allowing the cellphone to send messages about touch events on its touchscreen to the AR headset. The experiment software was developed using the Unity engine 2019.4.14f1c1.

**Task**. Each participant was seated on a chair. Participants were instructed to confirm the selection of a red target on the AR menu via tapping its counterpart on the cellphone touchscreen (see Figure 4b) as quickly and as accurately as possible.

**Experimental Design**. In order to make this pilot study isolate the confirmation selection accuracy of the HHSM technique, the AR menu only consists of one sub-menu. Specifically, the row and column numbers of the items on the cellphone touchscreen were the same as those in the AR menu, as shown in Figure 4b. We set the visual angle of a square item to $2.5^{\circ }$, which is the visual angle of the items in the “Start menu” of the Hololens 2 headset [19]. We explored various sub-menu sizes, including 1R2C, 2R1C, 1R3C, 3R1C, 2R2C, 2R3C, 3R2C, 3R3C, 3R4C, 4R3C, 4R4C, 4R5C, 5R4C, and 5R5C under the three holding gestures introduced in Section 3, respectively. The order of sub-menu size in conjunction with holding gesture was randomized for each participant. Under each condition, participants completed two blocks of trials, which consisted of a random permutation of selecting all the targets in a sub-menu.

**Results**. The results are shown in Figure 5. it was observed that the mean error rates surge dramatically when the sub-menu becomes more complex than 3R3C, while the mean completion times do not show a clear trend. Therefore, we concluded that 3R3C was the most complex sub-menu size for both confirmation techniques, i.e., the levels of the **Size** factor were 1R1C, 1R2C, 2R1C, 1R3C, 3R1C, 2R2C, 2R3C, 3R2C, and 3R3C. This finding will be useful in guiding the design of the following empirical studies.

## 5. Pilot Study for Selecting Optimal HHSM Instances

As the number of the **Size** factor’s levels was determined to be nine in Section 4, the design space derived in Section 3 delivered 54 (=9 size levels × 3 hold levels × 2 touch levels) HHSM instances. In this pilot study, we selected an optimal HHSM instance for each holding gesture.

**Participants.** We randomly recruited 18 participants (aged between 18 to 26) from a local university. Five among them were female and thirteen were male. They are all right-handed and six had AR experience.

**Apparatus.** The hardware setup was the same as that in Section 4. We set up our vertical AR menu as a 6R6C configuration, i.e., there are 36 icons distributed in a 6 × 6 grid in the menu. Then, the menu was compatible with all the nine sub-menu sizes mentioned above. The visual angle of a square item is the same as that in Section 4, i.e., $2.5^{\circ }$.

**Task.** Each participant was seated on a chair. Once an item on the AR menu turned red, the participant needed to first use their eye gaze to select the sub-menu that contains the red item and then confirm the selection of it via touch input on the cellphone. If a sub-menu was selected by eye gaze, the edges of all items in it become blue; see Figure 1. We required the participants to complete the trial as quickly and as accurately as possible.

**Experimental design.** We investigated the effects of the three dimensions of the HHSM technique’s design space, i.e., **Size**, **Hold**, and **Touch**, on menu selection performance in AR. The levels of the **Size** factor were the nine sizes determined in Section 4. The levels of the **Hold** factor were SH, OO, and TH. The levels of the **Touch** factor were tap and swipe (+ tap). We sampled 18 target items on the AR menu. To make them distributed evenly, we divided the whole AR menu into nine 2R2C sub-menus and then randomly selected two items from each sub-menu as the target items. Under each condition, participants had to complete two blocks of formal trials as previously described. Each block consisted of a random permutation of selecting the 18 target items. The levels of the factors were counterbalanced across the participants using balanced Latin square designs. To start a condition, a participant tapped anywhere on the cellphone touchscreen. At the beginning of each condition, there was a practice session where a participant needed to complete six trials. After the practice session, participants tapped anywhere on the cellphone touchscreen again to notify the system to start the timer and then began the formal trials. After participants completed the 36 trials (18 × 2), the timer stopped. Participants had a 30 s rest between every two conditions. The total experiment time excluding rest time was around one hour. Our experiment included a total of 36 trials × 9 size levels (1R1C, 1R2C, 2R1C, 1R3C, 3R1C, 2R2C, 2R3C, 3R2C, and 3R3C) × 3 hold levels (SH, OO, TH) × 2 touch levels (tap, swipe (+ tap)) × 18 participants = 34,992 trials. The dependent variables for each condition were: (i) the number of selection errors in 36 trials; and (ii) the mean completion time for selecting a target, which was calculated by dividing the total completion time under the condition recorded by the timer by the number of trials.

**Results.**Table 1 presents the means of the dependent variables for all 54 conditions.

**Qualitative Feedback.** For the levels of the sub-menu size factor, half of the participants (9/18) preferred 2R2C, while five preferred 2R3C. 1R1C, 1R3C, 3R2C, and 3R3C received only one vote each. P6’s comment was representative, “If the menu is not divided, it is hard to select a target accurately via eye gaze because the eye tracking is not very accurate. If the menu is divided into sub-menus with a size simpler than 2R2C, i.e., 1R2C, 2R1C, 1R3C, and 3R1C, the inaccurate eye tracking is only compensated for in one dimension because the sizes are anisotropic. If the sub-menus are too complex, there are too many distractors on the touchscreen to confirm selection accurately. 2R2C is isotropic and not too complex, which balances the above aspects well.” Under SH, participants (14/18) reported that it is not comfortable using the thumb of the hand holding the cellphone to frequently tap the items relatively far from the thumb; in contrast, participants preferred to use the swipe ( + tap) technique as they could perform the swipe from a comfortable position on the touchscreen. Under TH, most participants (16/18) mentioned that it is intuitive to use the left (or right) thumb to tap the two items on the left (or right) side. Participants did not show a dominant preference towards a touch technique under OO. Additionally, a participant suggested that the edges of the sub-menus should be displayed explicitly, which could facilitate users to quickly understand the sub-menu concept.

**Selecting optimal HHSM instances.** We adopted two-way repeated-measure ANOVA tests to analyze the effects of the **size** and **touch** factors on completion time under the three holding gestures, respectively. No significant difference was found. To limit the experiment time of this pilot study excluding rest time to around one hour, we had to make participants perform only 36 trials in each condition. In this case, just one selection error resulted in a 2.7% error rate. Hence, it makes little sense to select optimal HHSM instances based on error rate. Instead, we first selected several promising HHSM instances based on the number of errors in 36 trials (error number). Because all mean error numbers under the 54 conditions were larger than one, we set the standard of promising HHSM instances to “the mean error number is less than two”, and selected seven promising HHSM instances (see the mean error numbers enclosed by boxes in Table 1). Among them, three (SH-2R2C-Swipe, OO-2R2C-Swipe, and TH-2R2C-Tap) had the 2R2C sub-menu size. Also, the three instances were under SH, OO, and TH, respectively. Because of the advantages of 2R2C that received the most votes from participants (see the qualitative feedback part), we selected SH-2R2C-Swipe, OO-2R2C-Swipe, and TH-2R2C-Tap as the optimal HHSM instances for the SH, OO, and TH holding gestures, respectively.

## 6. User Study

In this user study, we compare the HHSM technique with three state-of-the-art techniques that also combine eye gaze and a hand-held device: *MAGIC touch* and *MAGIC tab* from the Look & Touch technique set [3], and *Eye + Device* from the Pinpointing technique set [1]; see Figure 6. All three techniques require the user to first move a pointer roughly to the proximity of the target item on a distant display via eye gaze. After that, the user could refine the pointer’s position before confirming the selection. The details of the refinement and confirmation phases of the techniques are as follows:

*MAGIC touch* (MTH). The user initiates the refinement phase by touching the touchscreen. This touch activates a circular selection mask with a fixed position on the distant display. Subsequently, the selection mask remains stationary and typically overlaps with the target item. Besides, the pointer no longer follows eye gaze. Then the user can use a finger to swipe on the touchscreen to move the pointer according to the finger’s relative movement from the initial touch position (relative positioning). If the user briefly touches a circular area on the device’s touchscreen, the pointer will jump to the corresponding position in the selection mask on the distant display (absolute positioning). Finally, the user confirms the selection by tapping a button at the bottom of the touchscreen.*MAGIC tab* (MTB). The user starts the refinement phase using the same method as *MAGIC touch*. Note that the selection mask remains stationary and typically overlaps with the target item. Then, the user cycles through the items that intersect the circular selection mask on the distant display using a horizontal swiping gesture on the device’s touchscreen. The user can also change the size of the selection mask with a vertical swiping gesture. The user confirms the selection by tapping the aforementioned button. In cases where the selection mask does not intersect with the target item after activation under both MAGIC Touch and MAGIC tab, the user can press a button located in the top portion of the touchscreen to cancel the selection mask.*Eye + Device* (ED). The user starts the refinement phase by pressing the device’s touchscreen with a finger. The user can then use the yaw and pitch of the device to finely control the pointer’s movements along the X and Y axes. The user confirms the selection by releasing the finger from the touchscreen.

**Participants**. We randomly recruited 24 participants (6 female and 18 male) from a local university, with ages ranging from 19 to 28. All participants were right-handed, and six of them had prior AR experience. The demographic information is shown in Table 2. However, it’s important to note that the AR experience of these six participants was limited to playing cellphone AR games, where virtual overlays are seen through the screen of a cellphone. This experience is not directly related to AR menu selection tasks using AR head-mounted displays, as in our experiment. Therefore, their prior AR experience did not have an impact on the results of our user study.

**Apparatus**. We developed the *MAGIC touch*, *MAGIC tab*, and *Eye + Device* techniques using the Unity engine, based on the apparatus described in Section 5.

**Task**. Participants were seated on a chair in front of the AR menu. Each trial started with a target item on the AR menu turning red to indicate it as the target. Participants were instructed to select the target item using eye gaze and a cellphone as quickly and as accurately as possible.

**Experimental design**. We examined the effects of two factors on user performance: the **hold** factor and the **technique** factor. The **hold** factor had three levels (SH, OO, and TH), while the **technique** factor had four levels (MTH, MTB, ED, and HHSM). For the relative positioning of MTH and the yaw and pitch gestures of ED, we invited all the participants to fine-tune their C/D ratios before the experiment. As a result, a finger translation of 1mm on the touchscreen made the cursor on the AR menu move $0.07^{\circ }$, while a cellphone rotation of $1^{\circ }$ made the AR cursor move $0.4^{\circ }$. Based on the findings in Section 5, the HHSM technique had three optimal instances, namely SH-2R2C-Swipe, OO-2R2C-Swipe, and TH-2R2C-Tap for the SH, OO, and TH levels, respectively. We adopted the suggestion from a participant in Section 5, i.e., we used dash white lines to visualize the edges of the sub-menus under HHSM. Each participant completed 12 sessions of trials, covering all possible combinations of the two factors. Each session consisted of six practice trials and two blocks of 18 formal trials using the same target items as in Section 5. The orders of the levels of the two factors were counterbalanced across participants using balanced Latin square designs. Participants took a five-minute rest after each session. The total experiment time excluding rest time was approximately 15 min per participant. The dependent variables for each session were the same as those in Section 5. This user study included a total of 36 trials × 4 technique levels (MTH, MTB, ED, and HHSM) × 3 hold levels (SH, OO, TH) × 24 participants = 10,368 trials.

After conducting the experiment, we asked the participants to rate on their experience using three tools: the NASA-TLX form [28], the measure of System Usability Scale (SUS) [29] and a questionnaire. The NASA-TLX form and the measure of System Usability Scale are used for assessing workload and perceived usefulness, respectively. We developed our questionnaire by selecting a set of typical questions from the comprehensive questionnaire in Khan et al.’s work [30]. The questionnaire contained six statements that were rated on a 5-point Likert scale, with 1 indicating strong disagreement and 5 indicating strong agreement. The six statements were as follows: (i) This technique is easy to learn. (Learnability). (ii) This technique adapts to various usage scenarios (Adaptability). (iii) This technique is easy to use (Ease of use). (iv) This technique is efficient (Efficient); (v) This technique effectively compensates for the problem of inaccurate eye tracking (Compensate). (vi) This is my favorite technique (Favorite). Besides, the experimenter engaged in one-on-one discussions with each participant and recorded their comments on their experience during the user study.

**Results**. Figure 7, Figure 8 and Figure 9 present the results. We used the non-parametric Aligned Rank Transformation (ART) test [31] to analyze the error number. We used two-way repeated-measures ANOVA with post hoc Tukey to analyze and completion time. We adopted Friedman tests with post hoc Wilcoxon Signed Rank tests to analyze the scores on the NASA-TLX form and the questionnaire.

**Error Number**. There was no significant interaction between the effects of the technique and hold factors ($F_{6,138}=1.431$, p=0.207). There was no significant difference between the levels of the technique factor ($F_{3,69}=1.243$, p=0.301). There was also no significant difference between the levels of the hold factor ($F_{2,46}=2.360$, p=0.106).

**Completion time**. There was a significant interaction between the effects of the technique and hold factors ($F_{6,138}=4.426$, p<0.001). A significant difference was found between the levels of the technique factor ($F_{3,69}=99.88$, p<0.001). A significant difference was also found between the levels of the hold factor ($F_{2,46}=14.38$, p<0.001). Below, we only report pairwise comparison results with statistical significance (p<0.05).

**Completion time under SH.** The completion time under HHSM (M = 1.14 s, SD = 0.10 s) was significantly shorter than those under MTH (M = 2.30 s, SD = 0.69 s, p<0.001), MTB (M = 2.97 s, SD = 0.95 s, p<0.001), and ED (M = 2.04 s, SD = 0.49 s, p<0.001). The completion time under MTB was significantly longer than those under MTH (p<0.001) and ED (p<0.001).

**Completion time under OO.** The completion time under HHSM (M = 1.12 s, SD = 0.12 s) was significantly shorter than those under MTH (M = 2.16 s, SD = 0.41 s, p<0.001), MTB (M = 2.71s, SD = 0.89s, p<0.001), and ED (M = 2.13 s, SD = 0.63 s, p<0.001). The completion time under MTB was significantly longer than those under MTH (p<0.001) and ED (p<0.001).

**Completion time under TH.** The completion time under HHSM (M = 0.93 s, SD = 0.12 s) was significantly shorter than those under MTH (M = 1.98 s, SD = 0.43 s, p<0.001), MTB (M = 2.23 s, SD = 0.46 s, p<0.001), and ED (M = 2.10 s, SD = 0.34 s, p<0.001).

**Workload**. We found a significant difference ($\chi^{2}\left(3\right)$=226.420, p< 0.001) in the workload score among the four conditions. The workload score under HHSM (M = 5.17, SD = 2.30) was significantly lower than those under MTH (M = 13.2, SD = 3.10, Z = −10.038, p< 0.001), MTB (M = 13.94, SD = 3.56, Z = −10.133, p< 0.001), and ED (M = 14.09, SD = 2.53, Z = −10.310, p< 0.001). The workload score under MTH (M = 13.4, SD = 3.10) was significantly lower than those under MTB (M = 14.07, SD = 3.08, Z = −3.235, p< 0.001) and ED (M = 14.26, SD = 2.26, Z = −2.995, p< 0.01).

**SUS**. We found a significant difference ($\chi^{2}\left(3\right)$=28.435, p< 0.001) in the SUS score among the four conditions. The SUS score under HHSM (M = 80, SD = 11.58) was significantly higher than those under MTH (M = 67.08, SD = 9.584, Z = −2.675, p< 0.01), MTB (M = 53.54, SD = 8.289, Z = −3.063, p< 0.01), and ED (M = 61.04, SD = 9.321, Z = −3.066, p< 0.01). The SUS score under MTB (M = 53.54, SD = 8.289) was significantly lower than those under MTH (M = 67.08, SD = 9.584, Z = −2.940, p< 0.01) and ED (M = 61.04, SD = 9.321, Z = −1.965, p< 0.05).

**Learnability**. We found a significant difference ($\chi^{2}\left(3\right)$=47.794, p< 0.001) in the Learnability score among the four conditions. The Learnability score under HHSM (M = 4.46, SD = 0.83) was significantly higher than those under MTH (M = 2.83, SD = 0.64, Z = −4.161, p< 0.001), MTB (M = 2.58, SD = 0.83, Z = −4.226, p< 0.001), and ED (M = 3.25, SD = 0.85, Z = −3.678, p< 0.001). The Learnability score under ED (M = 3.25, SD = 0.85) was significantly higher than those under MTH (M = 2.83, SD = 0.64, Z = −2.909, p< 0.01) and MTB (M = 2.58, SD = 0.83, Z = −2.5, p< 0.05). The Learnability score under MTH (M = 2.83, SD = 0.64) was significantly higher than those under MTB (M = 2.58, SD = 0.83, Z = −2.449, p< 0.05).

**Adaptability**. We found a significant difference ($\chi^{2}\left(3\right)$=17.958, p< 0.001) in the Adaptability score among the four conditions. The Adaptability score under HHSM (M = 4.17, SD = 0.7) was significantly higher than those under MTH (M = 3.25, SD = 1.07, Z = −3.152, p< 0.01), MTB (M = 3.08, SD = 1.02, Z = −2.943, p< 0.01), and ED (M = 3.33, SD = 0.96, Z = −2.699, p< 0.01).

**Ease of use**. We found a significant difference ($\chi^{2}\left(3\right)$=51.517, p< 0.001) in the Easy of use score among the four conditions. The Easy of use score under HHSM (M = 4.58, SD = 0.65) was significantly higher than those under MTH (M = 2.42, SD = 1.06, Z = −4.177, p< 0.001), MTB (M = 2.25, SD = 1.03, Z = −4.199, p< 0.001), and ED (M = 2.5, SD = 1.14, Z = −4.165, p< 0.001). The Easy of use score under MTH (M = 2.42, SD = 1.06) was significantly higher than those under MTB (M = 2.25, SD = 1.03, Z = −2.0, p< 0.05).

**Efficient**. We found a significant difference ($\chi^{2}\left(3\right)$=35.362, p< 0.001) in the Efficient score among the four conditions. The Efficient score under HHSM (M = 4.29, SD = 0.95) was significantly higher than those under MTH (M = 2.96, SD = 1.0, Z = −3.997, p< 0.001), MTB (M = 2.79, SD = 0.98, Z = −3.440, p< 0.001), and ED (M = 3.04, SD = 0.81, Z = −3.465, p< 0.001).

**Compensate**. We found a significant difference ($\chi^{2}\left(3\right)$=52.112, p< 0.001) in the Compensate score among the four conditions. The Compensate score under HHSM (M = 4.75, SD = 0.44) was significantly higher than those under MTH (M = 3.33, SD = 0.64, Z = −4.409, p< 0.001), MTB (M = 3.25, SD = 0.74, Z = −4.428, p< 0.001), and ED (M = 3.33, SD = 0.87, Z = −4.008, p< 0.001).

**Favorite**. We found a significant difference ($\chi^{2}\left(3\right)$=49.722, p< 0.001) in the Favorite score among the four conditions. The Favorite score under HHSM (M = 4.67, SD = 0.64) was significantly higher than those under MTH (M = 3.08, SD = 0.65, Z = −4.155, p< 0.001), MTB (M = 2.79, SD = 0.83, Z = −4.179, p< 0.001), and ED (M = 3.08, SD = 1.06, Z = −3.979, p< 0.001). The Favorite score under MTH (M = 3.08, SD = 0.65) was significantly higher than those under MTB (M = 2.79, SD = 0.83, Z = −2.121, p< 0.05).

**Learning curves**. The learning curves in Figure 9 show that the selection speed under the HHSM technique remains the most stable.

**Qualitative Feedback**. All participants preferred the HHSM technique the most because it strikes a good balance between accuracy and speed. Participant 5 (P5)’s comment was representative, stating, “*HHSM* only requires two simple steps. In the first step, the 2R2C sub-menus are large enough so that I can select the correct sub-menu via eye gaze easily. In the second step, it is very easy to select a target from four candidates in a 2R2C sub-menu via only one tap or swipe”. We summarize their comments for the other techniques as follows. Overall, although *MAGIC touch*, *MAGIC tab*, and ED were as accurate as HHSM, they were significantly more complex and slower. **Compared with HHSM that only requires two steps ((i) looking at the correct sub-menu and (ii) confirming the selection via a single tap/swipe),*****MAGIC touch*****and*****MAGIC tab*****require four steps, including (i) moving the gaze-driven pointer to the proximity of the target, (ii) activating the selection mask and fixing its location, (iii) refining the pointer’s location, and (iv) confirming the selection.** Under *MAGIC touch*, participants reported that absolute positioning was not accurate enough, and sometimes the pointer moved too fast or too slow when they performed relative positioning. Therefore, they often had to perform absolute or relative positioning several times to compensate for the problems. Under *MAGIC tab*, participants usually needed to cycle through several distractors to pre-select the target. ED also had the same problem of the relative positioning of *MAGIC touch*, i.e., the moving speed of the pointer for refinement was often not suitable.

**Discussion**. In step (ii) of MAGIC touch and MAGIC tab, a touch not only makes the selection mask appear but also fixes its location. Fixing the selection mask’s location is necessary, or the selection mask easily drifts due to inaccurate eye tracking. Also, the reasons why it often takes more than one touch action to refine the pointer’s position should be: (i) when using the absolute positioning of MAGIC touch, the fat finger problem would make the positioning inaccurate; (ii) when using the relative positioning of MAGIC touch, although the C/D ratio has been fine-tuned, it is not always suitable, which also would make the positioning inaccurate; and (iii) when using MAGIC tab, the tab order is unintuitive [3]. Additionally, while the *Eye + Device* technique does not have a selection mask, it also faces the C/D ratio issue in using yaw and pitch of the cellphone for pointer position refinement. Under HHSM, the 2R2C sub-menus are clearly indicated to users via white dash lines. The size of the 2R2C sub-menus ($5^{\circ }$) is proved to be large enough to effectively compensate for inaccurate eye tracking, ensuring that the sub-menu containing the target can keep being selected when confirming selection under HHSM. Thus, a position-fixing touch action like the one under *MAGIC touch* and *MAGIC tab* is not needed. Also, HHSM does not need a refinement step and only requires a single touch action for selection confirmation, which was found to be simple and easy for participants.

## 7. Limitations and Future Work

**Evaluating the HHSM technique in real-world settings**. The empirical studies in this work were conducted in an abstract style, i.e., all items in the menu were identical gray squares and the items did not bear icons. In each trial, the target item turns red to indicate it as the target. We designed the studies this way to eliminate the effects of the different colors and shapes of various icons on user performance. However, under real-world settings, all items in a menu bear icons. The target icon only turns its color after it is selected. In the future, we will conduct an ecological validity study to evaluate the HHSM technique under real-world settings.

**Combining HHSM with*****MAGIC touch*****for Non-Grid Interfaces.** HHSM is designed for real-world target selection in various grid menus such as those consisting of icons of applications, folders, files, and photos, etc. However, it is not suitable for interfaces where items are distributed randomly or in varying sizes. This is because the one-shot touch input designs of HHSM (the tap and swipe techniques) are only possible for grid menus, limiting the technique’s applicability. Nevertheless, by combining HHSM’s design essential of “dividing the interface into menu sections of the same size” with the absolute/relative positioning of *MAGIC touch*, we may achieve a better balance of accuracy and speed when dealing with non-grid interfaces where items are in mixed sizes. Under this approach (see Figure 10), the interface is divided into menu sections of the same size. To select a target, a user first looks at the menu section that intersects the target. They can then immediately start the refinement phase via touch input without a position-fixing touch action, as the one for fixing the position of the selection mask required by *MAGIC touch*/*MAGIC tab*. We plan to investigate this idea in future research.

**Investigating HHSM in the Mobile Scenario.** In this work, we investigated HHSM in the scenario where participants were sitting in a chair. In the future, we plan to investigate HHSM in the scenario where participants are walking. Due to oscillations caused by walking, we are interested in whether we need to enlarge the visual angle of sub-menus and users’ preferences for the holding gestures.

## 8. Conclusions

In this work, we propose the Hand-Held Sub-Menu (HHSM) technique, aiming at balancing accuracy and speed in gaze-touch grid menu selection in AR. Specifically, HHSM divides a grid menu into several sub-menus and maps the sub-menu pointed to by eye gaze onto the touchscreen of a hand-held device, making it possible to select a target via two easy selection steps. In this work, we first proposed the design space of HHSM. After that, we selected HHSM’s optimal instances for three holding gestures respectively through two pilot studies. By conducting a user study comparing HHSM and three state-of-the-art techniques, we found that there was no significant difference between them in selection error number and HHSM outperformed others in selection speed significantly.

## Figures and Tables

**Figure 1 sensors-23-09587-f001:**
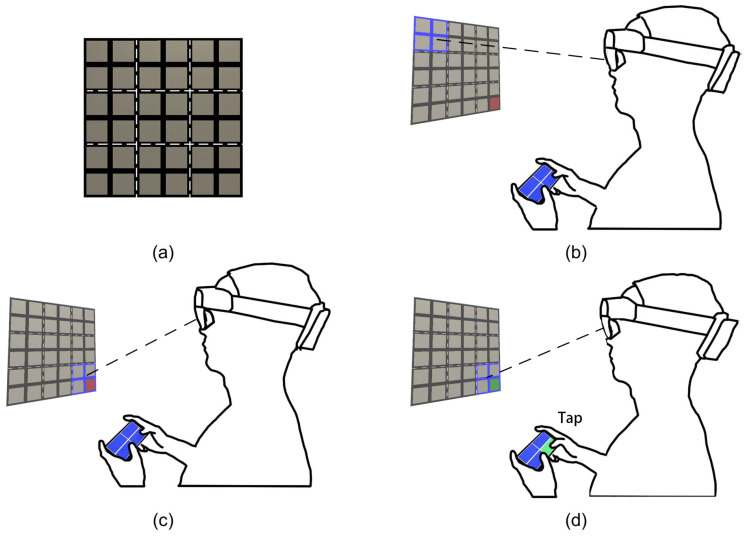
Illustration for an instance of the Hand-Held Sub-Menu (HHSM) technique. (**a**) The white dash lines divide a 6 × 6 grid menu in AR into nine 2 × 2 sub-menus, which are the candidate items for the user’s eye gaze. (**b**) The user’s eye gaze has selected the sub-menu at the upper-left corner, which is highlighted by coloring all its items’ edges in blue. (**c**) To select a target item, which is colored in red, the user first uses his eye gaze to select the sub-menu that contains the target item. (**d**) Then, the user taps the counterpart of the target item on the cellphone touchscreen to confirm the selection.

**Figure 2 sensors-23-09587-f002:**
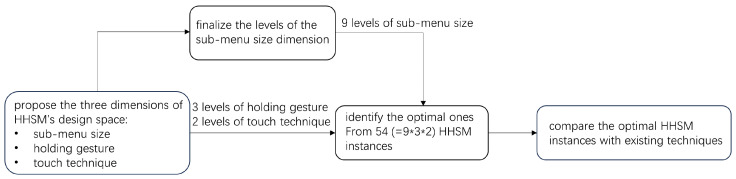
The research process of this work.

**Figure 3 sensors-23-09587-f003:**
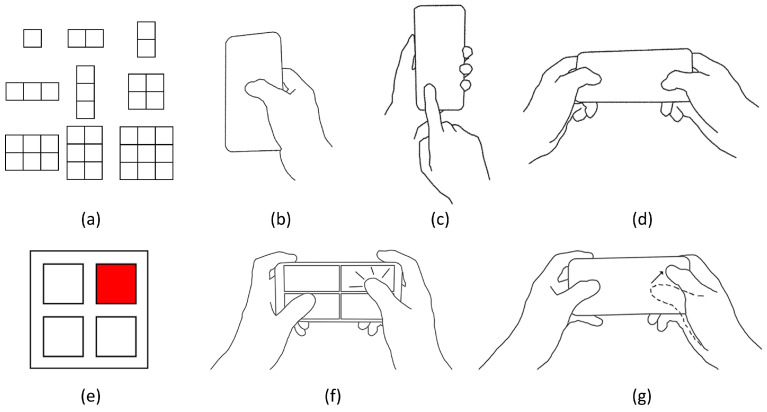
The three dimensions (**Size**, **Hold**, and **Touch**) of the HHSM technique’s design space. (**a**) Nine levels of the **Size** dimension: 1R1C, 1R2C, 2R1C, 1R3C, 3R1C, 2R2C, 2R3C, 3R2C, and 3R3C. Here, “R” stands for “rows of targets” and “C” stands for “columns of targets”. (**b**–**d**) Three levels of the **Hold** dimension. (**b**) SH (Single Hand), where the user uses a single hand to hold the device and interact with its touchscreen. (**c**) OO, where One hand holds the cellphone while the Other hand is used to interact with the touchscreen. (**d**) TH (Two Hands), where both hands are used to hold the two short edges of the cellphone, and both thumbs are used to interact with the touchscreen. (**e**) A 2 × 2 sub-menu where the upper-right red item is the target. (**f**,**g**) A user confirms the selection of the target using two levels of the **Touch** dimension respectively under TH. (**f**) The tap technique, which involves tapping the item’s counterpart on the cellphone touchscreen. (**g**) The swipe (+ tap) technique, which involves swiping on the touchscreen in the direction from the sub-menu center to the item.

**Figure 4 sensors-23-09587-f004:**
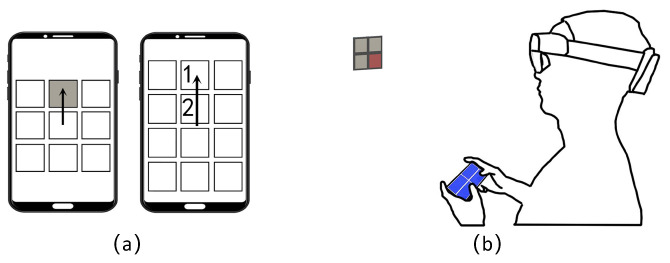
(**a**) Sub-menus more complex than 3R3C, which means nine items arranged in a 3 × 3 grid, are not suitable for the swipe (+ tap) technique. In the case of a 3R3C sub-menu (as shown on the left phone), only the grey item corresponds to the down-to-up swipe gesture. So, if the user swipes their finger from down to up on the touchscreen, the grey item will be selected. However, if the sub-menu is 4R3C (as shown on the right phone), which means twelve items arranged in a 4 × 3 grid, two items (target 1 and target 2) correspond to the down-to-up swipe gesture. (**b**) This image depicts the setup for the pilot study on sub-menu size. The AR menu consists of only one sub-menu.

**Figure 5 sensors-23-09587-f005:**
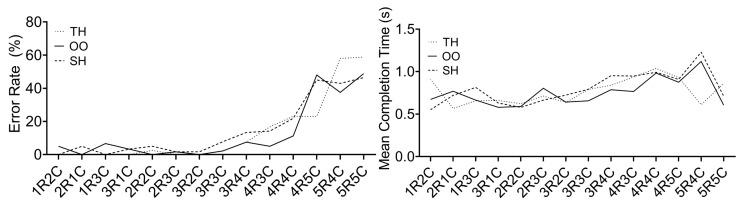
The (**left**,**right**) sub-figures illustrate the average error rates and completion times, respectively, that were obtained from the pilot study on sub-menu size.

**Figure 6 sensors-23-09587-f006:**
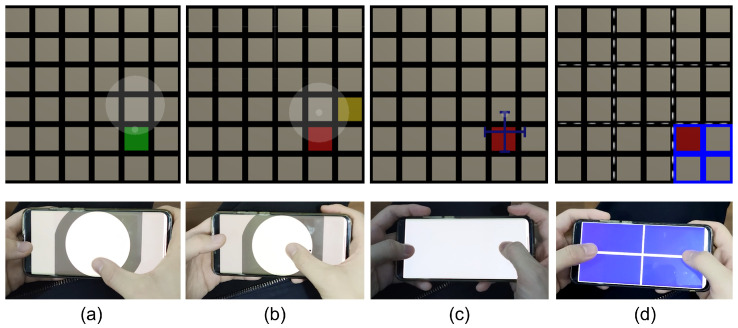
Four techniques evaluated in the formal user study when participants used two hands to hold and interact with a cellphone. The top and bottom rows show the AR menus and hand-device interactions under different techniques, respectively. (**a**) The *MAGIC touch* technique. [3]. (**b**) The *MAGIC tab* technique [3]. (**c**) The *Eye + Device* technique [1]. (**d**) The HHSM technique. At the beginning of each trial, the target item turns red. If it is selected successfully, it turns green. For the *MAGIC tab* technique, the items being cycled through turn yellow in turn.

**Figure 7 sensors-23-09587-f007:**
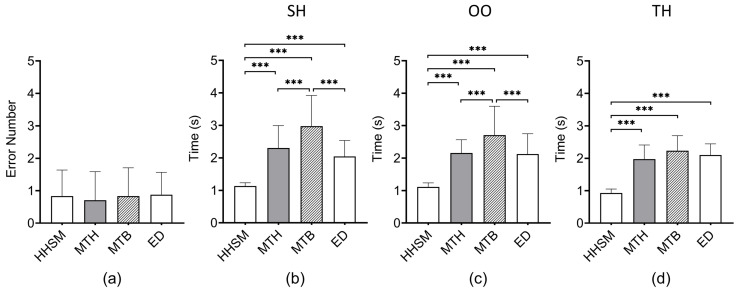
The results of user study 1. (**a**) The mean error number. (**b**–**d**) The mean selection times under SH, OO, and TH, respectively. *** means p<0.001.

**Figure 8 sensors-23-09587-f008:**
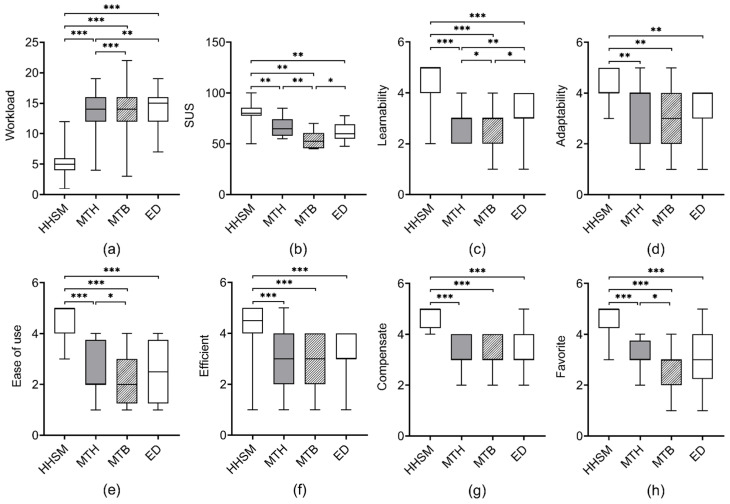
The * means p<0.05; ** means p<0.01; *** means p<0.001. results of user study 1. (**a**) The mean workload scores based on the NASA-TLX form. (**b**) The SUS (the measure of System Usability Scale) scores for measuring perceived usefulness. (**c**–**h**) The mean scores on six statements in our questionnaire.

**Figure 9 sensors-23-09587-f009:**
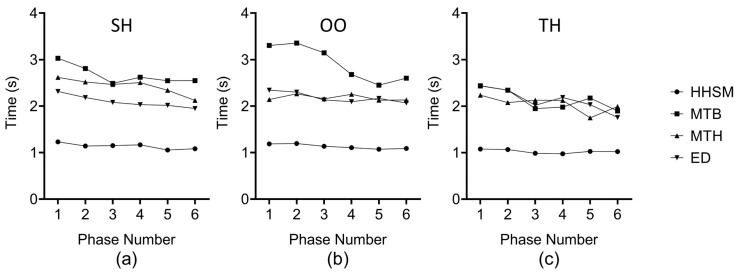
Each session of trials is divided into 6 phases for drawing the learning curve. (**a**–**c**) Learning curves of the four techniques under SH, OO, and TH, respectively.

**Figure 10 sensors-23-09587-f010:**
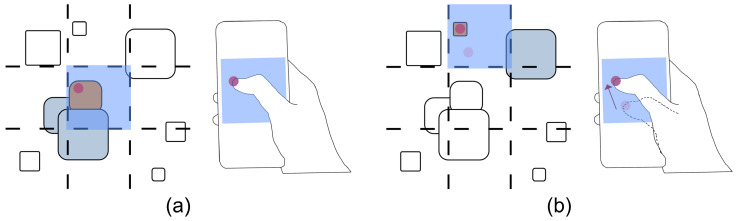
Illustration for combining a design essential of HHSM and the absolute/relative positioning of *MAGIC touch*. The dash lines divide the interface into sections of the same size in advance (pre-defined menu sections). To select a target (colored in orange), the user first looks at the section intersecting it (highlighted in blue). Then the user selects it via absolute positioning (**a**) or relative positioning (**b**) on a mobile device. The red dot represents the position where the user’s finger touches the touchscreen.

**Table 1 sensors-23-09587-t001:** The mean error number and completion times under 54 different conditions in the pilot study for selecting optimal HHSM instances. Each condition was a combination of three factors: hold (levels: SH, OO, and TH), size (levels: 1R1C, 1R2C, 2R1C, 1R3C, 3R1C, 2R2C, 2R3C, 3R2C, and 3R3C), and touch (levels: tap and swipe (+ tap)). The conditions with a mean error number smaller than two are enclosed by boxes.

SH			OO			TH		
	**Tap**	**Swipe (+ Tap)**		**Tap**	**Swipe (+ Tap)**		**Tap**	**Swipe (+ Tap)**
1R1C	4.0	3.2	1R1C	4.2	3.3	1R1C	3.7	4.5
	1.4 s	1.2 s		1.3 s	1.2 s		1.3 s	1.3 s
1R2C	2.3	1.4	1R2C	3.1	2.7	1R2C	3.1	2.7
	1.3 s	1.3 s		1.3 s	1.4 s		1.2 s	1.3 s
2R1C	2.6	2.3	2R1C	2.8	2.1	2R1C	1.6	1.8
	1.3 s	1.2 s		1.2 s	1.2 s		1.2 s	1.2 s
1R3C	3.2	3.2	1R3C	2.3	3.3	1R3C	4.2	2.7
	1.2 s	1.2 s		1.1 s	1.2 s		1.2 s	1.1 s
3R1C	3.0	2.5	3R1C	3.2	3.7	3R1C	2.2	2.7
	1.3 s	1.1 s		1.1 s	1.1 s		1.1 s	1.1 s
2R2C	2.2	1.9	2R2C	2.1	1.2	2R2C	1.4	2.0
	1.3 s	1.2 s		1.2 s	1.3 s		1.2 s	1.2 s
2R3C	2.4	5.7	2R3C	2.2	6.1	2R3C	2.3	3.0
	1.3 s	1.3 s		1.2 s	1.1 s		1.1 s	1.2 s
3R2C	4.2	3.4	3R2C	3.4	2.2	3R2C	1.8	3.2
	1.3 s	1.2 s		1.2 s	1.1 s		1.1 s	1.2 s
3R3C	4.7	4.3	3R3C	3.6	2.9	3R3C	2.8	3.9
	1.2 s	1.2 s		1.1 s	1.1 s		1.0 s	1.1 s

**Table 2 sensors-23-09587-t002:** The demographic information of participants in user study 1.

Variable	Group	Number of Participants	
		Count	Percentage
Gender	Female	6	25.00%
	Male	18	75.00%
Age	19 to 22 years	17	70.83%
	22 to 25 years	1	4.17%
	25 to 28 years	6	25.00%
Background	Undergraduate	17	70.83%
	Graduate student	7	29.17%
Preferred hand	Right hand	24	100.00%
	Left hand	0	0.00%
	Both hands	0	0.00%

## Data Availability

Data are contained within the article.

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
