# Peer review of "Balancing Accuracy and Speed in Gaze-Touch Grid Menu Selection in AR via Mapping Sub-Menus to a Hand-Held Device"

_sensors, 2023, doi:10.3390/s23239587_

Round 1
Reviewer 1 Report
Comments and Suggestions for Authors
The manuscript describes a proposed technique, the Hand-Held Sub-Menu (HHSM) technique, aimed at improving the balance between accuracy and speed in gaze-touch grid menu selection in augmented reality (AR) systems. The technique involves dividing the grid menu into sub-menus and utilizing a combination of eye gaze and touch on a hand-held device for more effective target selection.
The main findings suggest that HHSM can effectively address the issues related to selection accuracy and speed in AR systems. The study indicates an approximate error rate of 2%, with an average completion time per selection of about 0.83s for two-thumb interaction on the touchscreen, and 1.1s for single-finger interaction.
Following are the major comments which needs to be addressed before publication of this manuscript.
1. The abstract does not mention the number of participants involved in the empirical studies or their demographic characteristics. If the sample size was small or not diverse enough, the generalizability of the findings could be limited.
2. The manuscript highlights the completion time and error rate, but it may not delve into other important factors such as user preferences, learning curves, or potential usability issues that could affect the broader applicability and acceptance of the proposed technique. For reference authors may cite and get idea from the following papers where they have used important parameters including learnability, adaptability, ease-of-use, perceived usefulness etc.
i. Khan, I., & Khusro, S. (2020). Towards the design of context-aware adaptive user interfaces to minimize drivers’ distractions. Mobile Information Systems, 2020.
3. It has not specified whether the empirical studies were conducted in controlled environments or real-world settings. Without proper context, it is difficult to ascertain the practical viability of the HHSM technique in various AR usage scenarios.
4. Although the manuscript briefly mentions the need to balance accuracy and speed, it does not explicitly compare the HHSM technique with other existing techniques. Such comparisons are vital to understanding the relative advantages and limitations of the proposed approach in the context of the broader field
Comments on the Quality of English LanguageEnglish is fine.
Reviewer 2 Report
Comments and Suggestions for Authors
1. Line 158,Line160
What does “tangible proxy” here refer to? Pleases make a brief description or add a figure to demonstrate it.
2. Line 205
Five participants were recruited in this pilot experiment. Is the number large enough for data collection and statistical analysis? Please consider it and make explanations.
3. Line 230
In the experiment of section 4, only tap technique was experimented, then how do you draw the conclusion “…for both confirmation techniques”?
4. Line 323
In the refinement phase of the MAGIC tab technique, how can user move the selection mask vertically for that the vertical swiping gesture is set for size changing.
5. There’s a small mistake in Figure7’s description. The part (e) presents the favorite scores other than the workload scores.
6. In all of the three experiments, there were participants who had prior AR experience. Analysis of this factor should be considered and discussed.
Comments on the Quality of English Language
Minor editing of English language required.
Reviewer 3 Report
Comments and Suggestions for Authors
1. As mentioned in the size chapter, the final sizes are: 1R1C, 1R2C, 2R1C, 1R3C, 3R1C, 2R2C, 2R3C, 3R2C, 3R3C, 3R4C, 4R3C, 4R4C, 4R5C, 5R4C, 5R5C, etc. However, the size factor levels mentioned in the Pilot study on sub-menu size are: 1R1C, 1R2C, 2R1C, 1R3C, 3R1C, 2R2C, 2R3C, 3R2C and R3C. The article does not explain the reason for this inconsistency? Moreover, in the chapter Pilot study for selecting optimal HHSM instances, the menu is configured as 6R6C, which causes confusion for readers and makes them unclear about the context of the study.
2. Unable to clearly understand the entire research structure and process, so it is recommended to draw diagrams to assist explanation.
3. The data converted into pixels from the interface presentation size should be explained in detail so that the differences between different sub-menus can be clearly understood and compared.
4. Click reaction, why not use Fitt’s Law theory to test the subject’s operating performance in the AR environment?
